# Impact of Feeding Schedule on the Growth Performances of Tilapia, Common Carp, and Rice Yield in an Integrated Rice-Fish Farming System

**Md M. Billah [1], Md Kamal Uddin [1,\*], Mohd Y. A. Samad [1], Mohd Z. B. Hassan [2], Md Parvez Anwar [3], Isa Talukder [4], Md Shahjahan [4] and Ahmad Numery Ashfaqul Haque [1]**

[1]   Department of Land Management, Faculty of Agriculture, Universiti Putra Malaysia, Serdang, Selangor 43400, Malaysia; billahims@yahoo.com (M.M.B.); myusoffas@upm.edu.my (M.Y.A.S.); numerybau@gmail.com (A.N.A.H.)

[2]   Department of Aquaculture, Faculty of Agriculture, Universiti Putra Malaysia, Serdang, Selangor 43400, Malaysia; mzafri@upm.edu.my

[3]   Agro Innovation Laborator, Department of Agronomy, Faculty of Agriculture, Bangladesh Agricultural University, Mymensingh 2202, Bangladesh; parvezanwar@bau.edu.bd

[4]   Department of Fisheries Management, Faculty of Fisheries, Bangladesh Agricultural University, Mymensingh 2202, Bangladesh; isatalukder709@gmail.com (I.T.); mshahjahan75@gmail.com (M.S.)

\*   Correspondence: mkuddin07@gmail.com

**Abstract:** Feeding frequency, among various factors, greatly influences the production costs of aquaculture. In the present investigation, the effects of feeding schedule on the growth and production of tilapia and common carp were evaluated, along with rice yield, in an integrated rice-fish culture system. The experiment comprised 11 treatments, each with three replications, and a control treatment with no fish (T0). The fish in treatments T1–T5 were fed with rice bran once per week at 09:00 for T1, two days per week at 12:00 for T2, three days per week at 15:00 for T3, four days per week at 18:00 for T4, and five days per week at 09:00 and 18:00 for T5. The fish in treatments T6–T10 were fed an artificial floating feed with the same scheduled feeding frequency as T1–T5. The highest specific growth rate (SGR) in common carp (2.4%) and tilapia (4.3%) was found in T10. Similarly, the highest weight gains of tilapia (322.7 g) and common carp (180.9 g) were observed in T10 after 75 days of culture. In terms of rice, however, the highest recorded grain, straw, and biological yields of 5.6, 6.8, and 12.3 t ha$^{-1}$, respectively, were observed for control T0. Overall, the highest net return (USD 30,051 ha$^{-1}$) was found in T10. There was a 1504% greater net return and 98% higher benefit–cost ratio (BCR) in T10 compared to the control (T0). Five days of feeding per week at 09:00 and 18:00 was found to be the feeding schedule that resulted in the highest economic net return of those tested.

**Keywords:** rice; integrated rice-fish farming; feeding frequency; growth; yield

## 1. Introduction

Feed is usually the main factor that affects the cost of production in aquaculture facilities. Fish health depends on good water quality which, in turn, is strongly affected by the feeding schedule. Optimization of feeding strategies with adequate nutrients to meet fish requirements can optimize workload and feed waste, thus increasing profits [1]. Therefore, it is important to understand fish feeding schedules and rates for optimal growth and production. Some studies have revealed that daily feeding may not be necessary for maximum weight gain [2]. At the same time, a positive correlation has been found between weight gain and feeding frequency [3]. It has been reported that juvenile

Korean rockfish showed higher growth when fed every other day, rather than daily [4]. In addition, daily feeding of juvenile Korean rockfish has a relatively higher production cost [4].

It has been observed that insufficient feeding has negative impacts on growth in larvae due to their premature digestive tracts and high nutritional requirements [5]. Conversely, excessive feeding has resulted in poor growth of the fingerlings of *Catla catla*, *Labeo rohita*, and *Cirrhinus mrigala* in outdoor rearing systems [6]. The growth and health of fish has also been demonstrated to be compromised by underfeeding, leading to reduced survival rates [7,8]. Optimal feeding frequencies may increase growth by allowing food consumption during the reoccurrence of hungriness, resulting from gastrointestinal emptying events promoted by a regular feeding schedule. This may increase feeding efficiency, which then promotes growth [1]. The impact of feeding schedules on weight gain has been observed in several trials with hybrid sunfish *Lepomis cyanellus*, *Lepomis macrochirus*, yellow tail flounder *Limanda ferruginea*, Korean rockfish *Sebastes schlegelii*, zebrafish *Danio rerio*, dark barbel catfish *Pelteobagrus vachelli*, and cobia *Rachycentron canadum* [9–11].

A twice-daily feeding frequency has been recommended for hybrid tilapia *Oreochromis niloticus O. aureus*, while six times per day has also been effective for juveniles only [12–14]. However, feeding schedules are affected by differences in culture systems, genetic variation, age, diet, and fish [15]. It is important to establish a feeding frequency for the commonly cultured species that takes age, type of feed, and rearing conditions into account, among other factors [16,17]. Consequently, feeding frequency plays a vital role throughout the culture of fish. Additional investigations of feeding times have established that there should be an optimum feeding frequency in fish cultures [18]. However, few studies have investigated feeding frequencies for various fish in integrated rice-fish agriculture schemes [19]. Optimization of feeding frequencies in integrated rice-fish farming systems is currently lacking and requires evaluation. This study aimed to determine the effective fish-feed application periods for tilapia and common carp productivity as well as rice yield in an integrated rice-fish farming system.

## 2. Materials and Methods

### 2.1. Experimental Design

The experiment was carried out following a randomized complete block design. The experiment was conducted in 33 plots, each with an area of 15 $m^2$ ($5 \times 3$ m). The experiment comprised 11 treatments with three replicates each. The treatments and feeding schedules with rice bran were as follows: T0 (control)—no fish and no feed; T1—one day per week at 09:00; T2—two days per week at 12:00; T3—three days per week at 15:00; T4—four days per week at 18:00; T5—five days per week, with the same meal split into two feedings at 09:00 and 18:00. The fish in T6–T10 were fed via an artificial floating feed according to the same schedule as per T1–T5 (Table 1).

**Table 1.** Outline of research to evaluate the impact of fish-feeding schedules on the growth of tilapia and carp in the integrated rice-fish farming system.

| Treatments | Stocking Density $m^{-2}$ | Tilapia/Carp | Feedstuff | % of Body Weight | Application Day Week$^{-1}$ | Application Time |
|---|---|---|---|---|---|---|
| T0 | 6 | 1:1 | – | – | – | – |
| T1 | 6 | 1:1 | | 8 | 1 | M |
| T2 | 6 | 1:1 | Traditional | 8 | 2 | N |
| T3 | 6 | 1:1 | feed (rice | 8 | 3 | AN |
| T4 | 6 | 1:1 | bran) | 8 | 4 | E |
| T5 | 6 | 1:1 | | 8 | 5 | M + E |
| T6 | 6 | 1:1 | | 8 | 1 | M |
| T7 | 6 | 1:1 | Artificial | 8 | 2 | N |
| T8 | 6 | 1:1 | (floating | 8 | 3 | AN |
| T9 | 6 | 1:1 | feed) | 8 | 4 | E |
| T10 | 6 | 1:1 | | 8 | 5 | M + E |

M—morning (09:00); N—noon (12:00); AN—afternoon (15:00); E—evening (18:00). Treatments comprised five feeding frequencies in a week: one day (morning); two days (noon); three days (afternoon); four days (evening); and five days (morning and evening), respectively.

The stocking density of common carp, *C. carpio* L., with tilapia, *O. niloticus* (L.), was 6 m$^{-2}$ at a 1:1 ratio and fish were fed with food corresponding to 8% of their bodyweight (Table 1). In all treatments, feed was delivered manually in identical portions between 09:00 and 18:00, except for T5 and T10, in which feed was split into two identical portions provided in the morning and evening, at 09:00 and 18:00, respectively (Table 2).

**Table 2.** Details of feeding schedule of tilapia and carp in the integrated rice-fish farming system.

| Treatments | Description |
|------------|-------------|
| T0 | No feed and no fingerlings |
| T1 | Traditional feed (rice bran) + 1 day week$^{-1}$ in the morning |
| T2 | Traditional feed (rice bran) + 2 days week$^{-1}$ in the noon |
| T3 | Traditional feed (rice bran) + 3 days week$^{-1}$ in the afternoon |
| T4 | Traditional feed (rice bran) + 4 days week$^{-1}$ in the evening |
| T5 | Traditional feed (rice bran) + 5 days week$^{-1}$ in the morning and evening |
| T6 | Artificial (floating feed) + 1 day week$^{-1}$ in the morning |
| T7 | Artificial (floating feed) + 2 days week$^{-1}$ in the noon |
| T8 | Artificial (floating feed) + 3 days week$^{-1}$ in the afternoon |
| T9 | Artificial (floating feed) + 4 days week$^{-1}$ in the evening |
| T10 | Artificial (floating feed) + 5 days week$^{-1}$ in the morning and evening |

### 2.2. Growth and Yield of Fish

Individual fish weights were recorded from the individual experimental plots by random sampling [20]. After 75 days, growth parameters, such as weight gain, % weight gain, and specific growth rate (SGR), and survival were calculated for different temperature regimes using the following equations [21]:

Weight gain (g) = final weight (g) − initial weight (g)

$$Specific\ growth\ rate\ (SGR\%) = \frac{In\ final\ weight\ (g) - In\ initial\ weight\ (g)}{number\ of\ days} \times 100\% \tag{1}$$

$$Survival\ (\%) = \frac{Final\ no.\ of\ harvested\ fish}{Initial\ no.\ of\ fish} \times 100\% \tag{2}$$

Fish yield (kg ha$^{-1}$) = (Final weight − Initial weight) × Stocking density × Survival rate × Area.

### 2.3. Water Quality Parameters

Water temperature, pH, and dissolved oxygen (DO) concentration were measured in situ using a transportable pH meter and a polar graphic dissolved oxygen meter at 08:00 and 15:00 at 15-day intervals (DO and temperature: YSI model 58 dissolved-oxygen meter, YSI Co., Yellow Springs, OH, USA; pH: Hanna Instruments model HI 1270 pH probe, Hanna Instruments, Woonsocket, RI, USA). Free $CO_2$ (mgL$^{-1}$) and total alkalinity (mgL$^{-1}$) were measured using a titrimetric method using a phenolphthalein indicator with 0.0227N NaOH titrant and methyl orange indicator with 0.02N $H_2SO_4$ titrant, respectively. Levels of ammonia nitrogen (mgL$^{-1}$) and nitrite nitrogen (mgL$^{-1}$) were measured using a spectrophotometer (DR 1900, HACH, Elkhart, IN, USA). All tests were conducted following the Standard Methods Handbook [22].

### 2.4. Growth and Yield of Rice

Plant height (cm)
Normal plant height (cm) was recorded from arbitrarily nominated plants in every plot. Plant height was determined from the base to the tip of the uppermost spikelet of the plant and was stated in centimeters (cm) [23].

Number of total tillers per hill (total tillers·hill$^{-1}$)

To calculate the total tillers per hill, all tillers were counted from every sample, and an average taken. This comprised both effective and non-effective tillers [24].

Number of effective tillers per hill (effective tillers hill$^{-1}$)

To calculate the effective tillers per hill, only the ear-bearing tillers were calculated from every sample, and the average of samples was taken [25].

Number of grains per panicle (grains panicle$^{-1}$)

The numbers of grains per panicle of filled grains and unfilled grains were counted prior to the collection of samples.

Thousand-grain weight (g)

One thousand grains were taken arbitrarily from each plot, dried to 14% moisture content, and weighed using an electrical balance [24].

Grain yield (t ha$^{-1}$)

Grain yield was determined for each field by careful sun-drying and weighing grains. The weight of the sun-dried grains from each plot finally transformed into t ha$^{-1}$ [24].

Straw yield (t ha$^{-1}$)

The weight of the sun-dried straw was acquired from each unit plot that included straw. This yielded straw production per plot and was finally converted into t ha$^{-1}$ [26].

Biological yield (t ha$^{-1}$)

Grain production and straw production were taken together as biological production and calculated using the following formula:

Biological yield (t ha$^{-1}$) = Grain yield + straw yield [27].

Harvest Index (%)

$$\frac{Griain\ yield}{Biological\ yield} \times 100\% \tag{3}$$

The harvest index was calculated using grain production and biological production using the following formula:

Harvest Index (%) = [24]

## 2.5. Statistical Analysis

All data were subjected to one-way ANOVA using SAS 9.4 at $p \leq 0.05$ significance level and mean separations of experimental parameters by the LSD test.

## 3. Results

### 3.1. Growth Performances and Yields of Tilapia and Common Carp

The final body weight was observed to increase significantly ($p \leq 0.05$) with an increase in the feeding frequency from 1 to 5 days per week (Table 3). The common carp and tilapia that were fed five days per week in the morning (09:00) and evening (18:00) with artificial floating pellets (T10) had the highest SGR (2.4% and 4.3%, respectively) and survival rate (96.0% and 96.7%, respectively). The highest weight gains for common carp and tilapia were for T10 after 75 days of culture (322.7 and 180.9 g, respectively). The highest recorded yields of tilapia and common carp were also for T10 after 75 days of culture (4938.6 and 8809.7 kg ha$^{-1}$, respectively) ($p \leq 0.05$ in all cases).

**Table 3.** Growth performance and yield of tilapia and carp in the integrated rice-fish farming system for different feeding schedules.

| Treatments | Initial Weight (g) | | Final Weight (g) | | Weight Gain (g) | | SGR (%) | | Survival Rate (%) | | Yield (kg ha$^{-1}$) | |
|---|---|---|---|---|---|---|---|---|---|---|---|---|
| | Carp | Tilapia | Carp | Tilapia | Carp | Tilapia | Carp | Tilapia | Carp | Tilapia | Carp | Tilapia |
| T0 | No fish | No fish | – | – | – | – | – | – | – | – | – | – |
| T1 | 25.7 ± 0.5 | 12.5 ± 0.4 | 156.0 ± 4.0 [b] | 147.3 ± 3.3 [f] | 130.4 ± 4.0 [b] | 134.8 ± 3.3 [f] | 1.7 ± 1.2 [b] | 1.8 ± 1.1 [f] | 88.1 ± 4.0 [e] | 91.0 ± 2.0 [c] | 4123.1 [b] | 4021.3 [b] |
| T2 | 25.7 ± 0.5 | 12.5 ± 0.4 | 156.9 ± 0.9 [b] | 154.4 ± 3.5 [e,f] | 131.3 ± 0.9 [b] | 141.9 ± 3.5 [e] | 1.8 ± 1.4 [b] | 1.9 ± 1.1 [e] | 90.2 ± 0.2 [d] | 90.3 ± 0.3 [c] | 4245.7 [b] | 4182.7 [b] |
| T3 | 25.7 ± 0.5 | 12.5 ± 0.4 | 157.5 ± 1.5 [b] | 157.237.2 [e,f] | 131.8 ± 1.5 [b] | 144.7 ± 7.2 [e] | 1.8 ± 1.1 [b] | 1.9 ± 1.4 [e] | 93.3 ± 0.9 [a,b] | 90.9 ± 0.2 [c] | 4408.4 [ab] | 4286.8 [b] |
| T4 | 25.7 ± 0.5 | 12.5 ± 0.4 | 158.1 ± 2.1 [b] | 167.4 ± 3.4 [d,e] | 132.4 ± 2.1 [b] | 154.9 ± 3.4 [d] | 1.8 ± 1.2 [b] | 2.1 ± 1.1 [d] | 92.1 ± 0.1 [b] | 93.0 ± 0.1 [b] | 4368.3 [ab] | 4670.5 [b] |
| T5 | 25.7 ± 0.5 | 12.5 ± 0.4 | 159.0 ± 2.0 [b] | 175.6 ± 5.0 [d] | 133.3 ± 2.0 [b] | 163.1 ± 5.0 [d] | 1.8 ± 1.4 [b] | 2.2 ± 1.3 [d] | 95.8 ± 0.8 [a] | 94.8 ± 0.8 [b] | 4197.6 [b] | 4688.5 [b] |
| T6 | 25.7 ± 0.5 | 12.5 ± 0.4 | 204.1 ± 4.1 [a] | 292.3 ± 8.3 [c] | 178.5 ± 4.1 [a] | 279.8 ± 8.3 [c] | 2.4 ± 1.4 [a] | 3.7 ± 1.4 [c] | 92.1 ± 0.1 [b] | 92.7 ± 0.7 [b,c] | 4766.0 [a] | 7386.7 [a] |
| T7 | 25.7 ± 0.5 | 12.5 ± 0.4 | 205.0 ± 1.0 [a] | 313.2 ± 3.0 [b] | 179.3 ± 1.0 [a] | 300.7 ± 3.0 [b] | 2.4 ± 1.3 [a] | 4.0 ± 1.2 [b] | 93.9 ± 0.9 [a,b] | 93.6 ± 0.6 [b] | 4787.3 [a] | 7938.5 [a] |
| T8 | 25.7 ± 0.5 | 12.5 ± 0.4 | 205.2 ± 0.2 [a] | 319.4 ± 2.2 [a,b] | 179.6 ± 0.2 [a] | 306.9 ± 2.2 [b] | 2.4 ± 1.2 [a] | 4.1 ± 1.2 [b] | 94.7 ± 0.7 [a] | 93.4 ± 0.4 [b] | 4795.3 [a] | 8102.2 [a] |
| T9 | 25.7 ± 0.5 | 12.5 ± 0.4 | 205.6 ± 0.6 [a] | 330.7 ± 5.2 [a] | 179.9 ± 0.6 [a] | 318.1 ± 5.2 [a] | 2.4 ± 1.3 [a] | 4.2 ± 1.1 [a] | 94.9 ± 0.9 [a] | 94.9 ± 0.9 [b] | 4803.3 [a] | 8493.3 [a] |
| T10 | 25.7 ± 0.5 | 12.5 ± 0.4 | 206.5 ± 2.0 [a] | 335.2 ± 5.0 [a] | 180.9 ± 2.0 [a] | 322.7 ± 5.0 [a] | 2.4 ± 1.3 [a] | 4.3 ± 1.4 [a] | 96.0 ± 1.0 [a] | 96.7 ± 0.7 [a] | 4938.6 [a] | 8809.7 [a] |

Values represent means ± SD; different superscript letters within the same column denote significant differences at $p \leq 0.05$.

### 3.2. Water Quality Parameter during Culture Period

Though no significant differences were found for the water pH, our study revealed a substantial increase in ammonia (0.12–0.28) and nitrogen compounds (0.05–0.17), with an increase in the daily feeding schedule (Table 4).

**Table 4.** Water quality parameters in integrated rice-fish farming systems for different feeding schedules.

| Treatment | pH | Dissolved $O_2$ (mg $L^{-1}$) | Temperature (°C) | Alkalinity (mg$L^{-1}$) | $CO_2$ (ppm) | $NH_3$-$N_2$ (mg $L^{-1}$) | $NO_2$-$N_2$ (mg $L^{-1}$) |
|---|---|---|---|---|---|---|---|
| T0 | 7.2 ± 0.1 | 5.5 ± 0.1 [a] | 28.0 ± 0.9 [b,c,d] | 144.5 ± 3.7 [a] | 52.0 ± 1.9 [b] | 0.12 ± 0.02 [h] | 0.05 ± 0.01 [g] |
| T1 | 7.1 ± 0.1 | 5.4 ± 0.1 [a] | 28.8 ± 0.3 [a,b] | 120.0 ± 5.5 [e] | 40.5 ± 0.7 [e] | 0.20 ± 0.02 [f,g] | 0.07 ± 0.00 [f] |
| T2 | 7.1 ± 0.1 | 4.9 ± 0.1 [e] | 28.5 ± 0.4 [a,b,c] | 134.5 ± 2.5 [b] | 86.0 ± 1.1 [a] | 0.18 ± 0.01 [g] | 0.11 ± 0.01 [d] |
| T3 | 7.2 ± 0.1 | 5.2 ± 0.1 [b,c] | 28.3 ± 0.3 [a-d] | 104.0 ± 1.1 [f] | 35.0 ± 0.9 [f] | 0.20 ± 0.00 [f,g] | 0.14 ± 0.00 [b,c] |
| T4 | 7.1 ± 0.1 | 5.5 ± 0.1 [a] | 29.0 ± 0.9 [a] | 119.5 ± 2.2 [e] | 48.0 ± 0.4 [c] | 0.21 ± 0.00 [e,f] | 0.17 ± 0.01 [a] |
| T5 | 7.9 ± 0.2 | 5.1 ± 0.3 [d] | 27.8 ± 0.3 [cd] | 135.5 ± 2.1 [b] | 51.0 ± 1.2 [b] | 0.26 ± 0.02 [b,c] | 0.09 ± 0.00 [e] |
| T6 | 7.4 ± 0.1 | 5.3 ± 0.1 [b,c] | 28.3 ± 0.9 [ab] | 129.0 ± 4.7 [c] | 48.0 ± 1.4 [c] | 0.23 ± 0.02 [d,e] | 0.14 ± 0.01 [b,c] |
| T7 | 7.3 ± 0.1 | 5.3 ± 0.1 [b] | 29.0 ± 0.2 [a] | 127.5 ± 2.8 [cd] | 49.5 ± 2.3 [b,c] | 0.27 ± 0.00 [a,b,c] | 0.13 ± 0.01 [c] |
| T8 | 7.3 ± 0.1 | 5.2 ± 0.1 [c,d] | 28.0 ± 0.0 [b,c,d] | 122.5 ± 1.2 [d,e] | 44.0 ± 0.7 [d] | 0.29 ± 0.01 [a] | 0.18 ± 0.02 [a] |
| T9 | 7.2 ± 0.1 | 5.3 ± 0.2 [b] | 27.8 ± 0.4 [c,d] | 118.5 ± 0.3 [e] | 52.0 ± 2.2 [b] | 0.25 ± 0.01 [c,d] | 0.15 ± 0.00 [b] |
| T10 | 7.2 ± 0.1 | 5.3 ± 0.1 [b] | 27.5 ± 0.6 [d] | 117.3 ± 3.5 [e] | 51.3 ± 1.3 [b] | 0.28 ± 0.02 [a,b] | 0.17 ± 0.01 [a] |

Values represent means ± SD; different superscript letters within the same column denote significant differences at $p \leq 0.05$.

### 3.3. Plant Height of Rice (BRRI dhan29)

Seventy-five days after transplantation, plant height was significantly higher in T0, T10, and T9 (109.2, 108.7, and 108.7 cm, respectively). We observed that 30 days after transplanting, T0 was higher than all other treatments; this trend continued to 45 days post-transplantation. The control was significantly taller than all treatments 30 and 45 days after transplanting. The greatest decrease in plant height was 0.5% for T10, followed by 1.4% at T8 and 3.2% at T5, as compared to T0 (Table 5).

**Table 5.** Plant height (cm) of rice (BRRI dhan29) in the rice-fish farming system at various days post-transplantation for different feeding schedules.

| Treatments | Plant Height (cm) at Various Days Post-Transplantation | | | |
|---|---|---|---|---|
| | 30 Days | 45 Days | 60 Days | 75 Days |
| T0 | 83.7 ± 5.6 [a] | 87.7 ± 2.5 [a] | 98.1 ± 3.2 [a] | 109.2 ± 7.1 [a] |
| T1 | 75.0 ± 1.5 [c] | 74.1 ± 4.2 [d] | 89.3 ± 4.7 [c] | 97.7 ± 2.0 [e] |
| T2 | 76.8 ± 2.2 [c] | 74.7 ± 0.9 [d] | 90.0 ± 2.0 [c] | 99.7 ± 2.4 [d] |
| T3 | 78.8 ± 2.0 [b,c] | 79.1 ± 4.7 [c] | 91.8 ± 3.3 [c] | 105.7 ± 3.4 [b] |
| T4 | 78.4 ± 2.4 [b,c] | 79.0 ± 2.7 [c] | 91.7 ± 0.7 [c] | 103.7 ± 4.0 [c] |
| T5 | 79.2 ± 1.7 [b] | 80.8 ± 7.1 [c] | 92.8 ± 1.6 [b,c] | 106.2 ± 1.2 [ab] |
| T6 | 77.3 ± 1.8 [c] | 76.9 ± 1.0 [cd] | 90.2 ± 0.2 [c] | 101.7 ± 1.8 [c] |
| T7 | 77.67 ± 1.45 [c] | 77.5 ± 2.7 [b,c,d] | 91.1 ± 2.7 [c] | 102.3 ± 3.9 [c] |
| T8 | 79.3 ± 1.5 [b] | 81.2 ± 5.6 [b,c] | 93.6 ± 2.5 [b,c] | 107.7 ± 1.0 [a] |
| T9 | 81.1 ± 1.7 [ab] | 83.3 ± 1.7 [b] | 95.9 ± 1.8 [b] | 108.7 ± 3.7 [a] |
| T10 | 79.3 ± 1.2 [b] | 82.1 ± 4.7 [b] | 94.1 ± 3.8 [b] | 108.7 ± 3.8 [a] |

Values represent means ± SD; different superscript letters within the same column denote significant differences at $p \leq 0.05$.

### 3.4. Tillers Number Per Hill of Rice (BRRI dhan29)

The control T0 showed the highest number of tillers hill$^{-1}$ (12.7), as well as significantly higher number of tillers·hill$^{-1}$ than all other treatments, at both 30 and 75 days after transplanting. By contrast, the control T0 did not have significantly more tillers than T08, T09, and T10 (10.8, 11.7, and T11.3, respectively) 45 days after transplanting. This was also the case 60 days after transplanting. At 75 days after transplanting, however, the number of total tillers·hill$^{-1}$ for T08, T09, and T10 decreased by 14.1%, 3.6%, and 8.8%, respectively, compared to T0 (Table 6).

**Table 6.** Total number of tillers hill$^{-1}$ of rice (BRRI dhan29) in the rice-fish farming system at various days post transplantation for different feeding schedules.

| Treatments | Number of Total Tillers Per Hill at Various Days Post-Transplantation | | | |
|---|---|---|---|---|
| | 30 Days | 45 Days | 60 Days | 75 Days |
| T0 | 11.1 ± 0.2 [a] | 11.3 ± 0.6 [a] | 11.8 ± 0.8 [a] | 12.7 ± 0.3 [a] |
| T1 | 8.3 ± 0.6 [f] | 8.9 ± 0.5 [d] | 9.1 ± 0.1 [e] | 9.3 ± 0.6 [e] |
| T2 | 8.8 ± 0.2 [e] | 9.0 ± 0.2 [d] | 9.1 ± 0.2 [e] | 9.4 ± 0.8 [e] |
| T3 | 9.6 ± 0.7 [c] | 9.9 ± 0.2 [b,c] | 10.2 ± 0.2 [b] | 10.6 ± 0.7 [c] |
| T4 | 9.4 ± 0.5 [c] | 9.5 ± 0.4 [c,d] | 9.7 ± 0.9 [c] | 10.0 ± 0.5 [d] |
| T5 | 9.7 ± 0.3 [b,c] | 10.3 ± 0.3 [b] | 10.6 ± 0.5 [b] | 10.8 ± 0.2 [c] |
| T6 | 8.9 ± 0.5 [e] | 9.2 ± 0.4 [d] | 9.3 ± 0.3 [d] | 9.9 ± 0.5 [e] |
| T7 | 9.0 ± 0.3 [d] | 9.3 ± 0.6 [c,d] | 9.6 ± 0.4 [c] | 10.0 ± 0.7 [e] |
| T8 | 9.9 ± 0.7 [b] | 10.8 ± 0.7 [a,b] | 11.0 ± 0.3 [a,b] | 10.9 ± 0.2 [c] |
| T9 | 10.4 ± 0.2 [b] | 11.2 ± 0.7 [a] | 11.7 ± 0.7 [a] | 12.2 ± 0.2 [a] |
| T10 | 9.9 ± 0.2 [b] | 11.1 ± 1.0 [a] | 11.3 ± 0.1 [a] | 11.6 ± 0.4 [b] |

Values represent means ± SD; different superscript letters within the same column denote significant differences at $p \leq 0.05$.

### 3.5. Yield and Yield Contributing Characteristics of Rice (BRRI dhan29)

The number of effective tillers per hill, number of non-effective tillers per hill, number of grains·panicle$^{-1}$, number of sterile spikelets·spike$^{-1}$, straw yield (t ha$^{-1}$), grain yield (t ha$^{-1}$), biological yield (t ha$^{-1}$), and harvest index (%) were significantly affected by the fish feeding schedule (Table 7). The highest grain yield (5.6 t ha$^{-1}$) and straw yield (6.8 t ha$^{-1}$) were recorded for T0. The biological yield was highest (12.3 ha$^{-1}$) in the control (T0) and lowest (8.1 ha$^{-1}$) in T1. Similarly, the harvest index was highest (44.75%) in T0 and lowest (42.8%) in T1.

### 3.6. Economic Evaluation of Different Feeding Schedules of Tilapia and Common Carp in the Integrated Rice-Fish Farming System

The net return from rice-fish culture was higher than only rice culture in the control (T0) when compared with the revenue generated from the plot of rice-fish farming due to the presence of fish (Table 8). Our results showed that fish fed five days per week, twice per day in the morning and evening, had the highest net return in T10 (USD 30,051 ha$^{-1}$). T10 also provided 1504% greater net returns than the control (T0). T10 gave 98% higher BCR than T0 (Table 8).

**Table 7.** Yield and yield contributing characteristics of rice (BRRI dhan29) in the rice-fish farming system for different feeding schedules.

| Treatments | No. of Effective Tillers·Hill$^{-1}$ | No. of Non-Effective Tillers·Hill$^{-1}$ | No. Grains·Panicle$^{-1}$ | No. of Sterile Spikelets·Spike$^{-1}$ | 1000-Grain wt. (g) | Grain Yield (t ha$^{-1}$) | Straw Yield (t ha$^{-1}$) | Biological Yield (t ha$^{-1}$) | Harvest Index (%) |
|---|---|---|---|---|---|---|---|---|---|
| T0 | 12.7 ± 2.7 [a] | 0.9 ± 0.1 [f] | 132.9 ± 13.0 [a] | 5.0 ± 1.0 [f] | 17.1 ± 1.1 [a] | 5.5 ± 0.5 [a] | 6.8 ± 0.2 [a] | 12.3 ± 2.7 [a] | 44.8 ± 1.5 [a] |
| T1 | 8.4 ± 0.4 [e] | 1.5 ± 0. 2 [a] | 100.3 ± 0.3 [f] | 7.4 ± 0.4 [a] | 14.1 ± 0.1 [e] | 3.5 ± 0.1 [d] | 4.6 ± 0.2 [d] | 8.1 ± 0.3 [f] | 42.8 ± 0.9 [c] |
| T2 | 8.6 ± 0.1 [e] | 1.4 ± 0.1 [ab] | 103.1 ± 2.0 [e] | 7.3 ± 0.3 [a] | 14.2 ± 0.2 [d] | 3.7 ± 0.1 [d] | 4.9 ± 0.1 [d] | 8.6 ± 0.1 [e] | 43.1 ± 0.7 [b] |
| T3 | 10.5 ± 0.5 [c] | 1.2 ± 0.1 [b] | 118.8 ± 0.8 [b] | 6.3 ± 0.3 [c] | 15.5 ± 0.5 [c] | 4.6 ± 0.1 [b] | 5.9 ± 0.1 [c] | 10.5 ± 0.1 [b] | 44.0 ± 0.8 [a] |
| T4 | 9.7 ± 0.1 [d] | 1.3 ± 0.1 [b] | 114.7 ± 0.7 [c] | 6.6 ± 0.6 [bc] | 15.3 ± 0.3 [c] | 4.3 ± 0.3 [b] | 5.6 ± 0.2 [c] | 9.9 ± 0.2 [c] | 43.6 ± 0.7 [b] |
| T5 | 10.7 ± 0.1 [c] | 1.2 ± 0.3 [b] | 125.5 ± 0.5 [ab] | 6.2 ± 0.2 [d] | 15.7 ± 0.7 [c] | 4.9 ± 0.2 [b] | 6.1 ± 0.1 [b] | 11.0 ± 0.1 [ab] | 44.3 ± 0.6 [a] |
| T6 | 9.0 ± 0.2 [d] | 1.3 ± 0.3 [ab] | 105.8 ± 1.0 [e] | 7.0 ± 0.2 [b] | 14.5 ± 0.5 [d] | 4.0 ± 0.3 [d] | 5.2 ± 0.2 [c] | 9.2 ± 0.2 [d] | 43.2 ± 1.3 [b] |
| T7 | 9.37 ± 0.37 [d] | 1.3 ± 0.2 [ab] | 108.4 ± 2.0 [d] | 6.9 ± 0.1 [b] | 14.9 ± 0.1 [d] | 4.1 ± 0.1 [c] | 5.4 ± 0.4 [c] | 9.5 ± 0.3 [d] | 43.3 ± 2.3 [b] |
| T8 | 11.11 ± 0.11 [b] | 1.1 ± 0.1 [c] | 128.3 ± 0.3 [a] | 5.9 ± 0.1 [e] | 16.0 ± 0.5 [bc] | 5.2 ± 0.2 [a] | 6.5 ± 0.5 [ab] | 11.6 ± 0.6 [ab] | 44.3 ± 1.6 [a] |
| T9 | 12.1 ± 0.1 [ab] | 1.0 ± 0.3 [c] | 131.4 ± 1.4 [a] | 5.3 ± 0.1 [f] | 16.4 ± 0.4 [b] | 5.4 ± 0.1 [a] | 6.7 ± 0.1 [a] | 12.1 ± 0.2 [a] | 44.6 ± 0.1 [a] |
| T10 | 11.5 ± 0.1 [ab] | 1.1 ± 0.1 [d] | 129.8 ± 1.5 [a] | 5.7 ± 0.3 [e] | 16.2 ± 0.2 [b] | 5.3 ± 0.1 [a] | 6.6 ± 0.2 [a] | 12.0 ± 0.2 [ab] | 44.6 ± 1.1 [a] |

Values represent means ± SD; different superscript letters within the same column denote significant differences at *p* ≤ 0.05.

**Table 8.** Costs and economic return based on yield of tilapia and common carp with rice (*BRRI dhan29*) in the integrated rice-fish farming system in Bangladesh. Currency in USD ($).

| Cost Items | T0 ($ ha$^{-1}$) | T1 ($ ha$^{-1}$) | T2 ($ ha$^{-1}$) | T3 ($ ha$^{-1}$) | T4 ($ ha$^{-1}$) | T5 ($ ha$^{-1}$) | T6 ($ ha$^{-1}$) | T7 ($ ha$^{-1}$) | T8 ($ ha$^{-1}$) | T9 ($ ha$^{-1}$) | T10 ($ ha$^{-1}$) |
|---|---|---|---|---|---|---|---|---|---|---|---|
| **A: Variable costs** | | | | | | | | | | | |
| Land preparation and dike(*ail*) production | 1031 | 1031 | 1031 | 1031 | 1031 | 1031 | 1031 | 1031 | 1031 | 1031 | 1031 |
| Rice seed | 94 | 94 | 94 | 94 | 94 | 94 | 94 | 94 | 94 | 94 | 94 |
| Seed sprouting of rice | 100 | 100 | 100 | 100 | 100 | 100 | 100 | 100 | 100 | 100 | 100 |
| Irrigation | 150 | 150 | 150 | 150 | 150 | 150 | 150 | 150 | 150 | 150 | 150 |
| Weeding | 50 | 50 | 50 | 50 | 50 | 50 | 50 | 50 | 50 | 50 | 50 |
| Artificial floating fish feed | – | – | – | – | – | – | 1615 | 1615 | 1615 | 1615 | 1615 |
| Rice bran | – | 241 | 241 | 241 | 241 | 241 | – | – | – | – | – |
| Tilapia fingerlings ($0.03/piece) | – | 900 | 900 | 900 | 900 | 900 | 900 | 900 | 900 | 900 | 900 |
| Common carp fingerlings ($0.06/piece) | – | 1800 | 1800 | 1800 | 1800 | 1800 | 1800 | 1800 | 1800 | 1800 | 1800 |

**Table 8.** *Cont.*

| Cost Items | T0 ($ ha$^{-1}$) | T1 ($ ha$^{-1}$) | T2 ($ ha$^{-1}$) | T3 ($ ha$^{-1}$) | T4 ($ ha$^{-1}$) | T5 ($ ha$^{-1}$) | T6 ($ ha$^{-1}$) | T7 ($ ha$^{-1}$) | T8 ($ ha$^{-1}$) | T9 ($ ha$^{-1}$) | T10 ($ ha$^{-1}$) |
|---|---|---|---|---|---|---|---|---|---|---|---|
| Hired labor for feeding and harvesting fish and rice | – | 163 | 163 | 163 | 163 | 163 | 163 | 163 | 163 | 163 | 163 |
| Post-harvest operation | 188 | 188 | 188 | 188 | 188 | 188 | 188 | 188 | 188 | 188 | 188 |
| Miscellaneous | 25 | 25 | 25 | 25 | 25 | 25 | 25 | 25 | 25 | 25 | 25 |
| **Subtotal of variable costs** | 1638 | 4742 | 4742 | 4742 | 4742 | 4742 | 6116 | 6116 | 6116 | 6116 | 6116 |
| **B: Fixed costs** | | | | | | | | | | | |
| Protection fence by net cover | 100 | 100 | 100 | 100 | 100 | 100 | 100 | 100 | 100 | 100 | 100 |
| TSP | – | 47 | 47 | 47 | 47 | 47 | 47 | 47 | 47 | 47 | 47 |
| Cow dung | – | 15 | 15 | 15 | 15 | 15 | 15 | 15 | 15 | 15 | 15 |
| Urea | – | 62 | 62 | 62 | 62 | 62 | 62 | 62 | 62 | 62 | 62 |
| **Subtotal of fixed costs:** | 100 | 224 | 224 | 224 | 224 | 224 | 224 | 224 | 224 | 224 | 224 |
| **Total Cost (A+B):** | 1738 | 4966 | 4966 | 4966 | 4966 | 4966 | 6340 | 6340 | 6340 | 6340 | 6340 |
| Tilapia ($1.85/kg) | – | 7439 | 7738 | 7931 | 8640 | 9239 | 13,665 | 14,686 | 14,989 | 15,713 | 16,298 |
| Common carp ($3.58/kg) | – | 14,761 | 15,200 | 15,782 | 15,639 | 15,027 | 17,062 | 17,139 | 17,167 | 17,196 | 17,680 |
| **Total fish income ($/ha)** | – | 22,200 | 22,938 | 23,713 | 24,279 | 24,266 | 30,728 | 31,825 | 32,156 | 32,908 | 33,978 |
| Grain yield ($370.38 t$^{-1}$) | 2063 | 1301 | 1388 | 1733 | 1624 | 1830 | 1489 | 1538 | 1931 | 2025 | 1999 |
| Straw yield ($61.73 t$^{-1}$) | 424 | 290 | 306 | 367 | 350 | 384 | 327 | 336 | 404 | 419 | 414 |
| **Total rice income ($/ha)** | 2487 | 1591 | 1693 | 2099 | 1974 | 2214 | 1816 | 1873 | 2336 | 2444 | 2413 |
| **Gross return (GR) $** | 2487 | 23,791 | 24,631 | 25,812 | 26,253 | 26,480 | 32,544 | 33,698 | 34,492 | 35,352 | 36,391 |
| **Net return $ (NR = GR − TC)** | 749 | 18,825 | 19,665 | 20,846 | 21,287 | 21,514 | 26,204 | 27,358 | 28,152 | 29,012 | 30,051 |
| **BCR** | **1.4** | **3.2** | **3.3** | **3.5** | **3.5** | **3.6** | **3.4** | **3.5** | **3.6** | **3.7** | **3.8** |

## 4. Discussion

Feeding schedules were observed to have a significant impact on plant height, tiller number, and growth performances of tilapia and carp *(p ≤ 0.05)*. Rice yield, straw yield, plant height, and tiller number were generally high across the entire feeding schedule, which may be due to increased nutrient availability [28]. Similarly, fish also benefited from epiphytic and benthic food as well as the shade of rice plants that maintained favorable water temperatures during hot summer days [29–31]. Although T0 had the highest rice yield and straw yield, the maximum feeding schedule of five days per week also showed a high rice yield and straw yield, which indicates that the integrated system was a favorable environment for rice production [32].

Feeding frequency also had a significant effect on the growth performance of carp and tilapia. A feeding frequency of five days per week (morning and evening) was the most optimal of those tested, overall, for the rice-fish integrated farming system examined in this study. This study showed that feeding via an artificial floating feed five days per week in the morning (09:00) and evening (18:00) significantly affected the final body weight gain and SGR of common carp and tilapia. The final body weight in this treatment was higher than that in fish fed two, three, or four times per week. This observation is in agreement with previous studies that determined the optimal feeding frequency of tilapia [2,15]. Interestingly, the feeding frequency showed significant differences for diet-fed tilapia but not common carp. This may be due to the naturally available food for common carp found in all rice fields. Persian sturgeons fed four and five times per day gained more weight than when fed just three meals [1,33]. Likewise, the feeding times per day and fasting at the end of the week impacted weight gains in rainbow trout (*Oncorhynchus mykiss*) and showed that feeding for seven days is essential for best weight gain [34,35]. These outcomes do not concur with the observations of De Silva and Anderson [16], who reported that extreme feeding does not affect weight gain. A significantly higher weight increase occurred in hybrid tilapia fed multiple times in contrast with those fed twice daily [13]. An extraordinary high feeding frequency was observed to be optimal for sturgeon fed 24 meals daily using a programmed feeder [36]. Ideal feeding frequencies have also been likewise for different species, for example, the dark barbel catfish (twice per day), yellow tail fumble (once per day), Korean rockfish (three times per day), sunfish, and adolescent yellow croaker (eight times per day) [37,38]. Therefore, from an economic perspective, a suitable feeding procedure is a basic framework for any successful aquaculture endeavor.

Most of the water quality parameters in the present study were within suitable ranges for the growth of fish. The accessibility of $CO_2$ for phytoplankton development was linked to alkalinity; normal pH created an appropriate amount of $CO_2$ for plankton generation [39]. Most feed nutrients consumed by fish in feed-based aquaculture were discharged into the surroundings. Approximately one-third of the nutrients in the feed are utilized by fish [40]. Nevertheless, higher feeding schedules may generate additional waste, as observed in the five-day-per-week feeding in morning and evening [41].

We found that the highest net return from the rice-fish culture was in fish fed five days per week, twice per day in the morning and evening in T10, which gave 98% higher BCR. Although the rice yield was lower than that of the rice monoculture, the total return for the rice-fish system was significantly higher. Thus, rice yield loss was outweighed by the higher return from fish under the examined rice-fish system. The resultant increase in gross margins for the rice-fish system results in a benefit–cost ratio of 2.9. Therefore, at the farm level, the use of an integrated rice-fish system appears to be an economically viable alternative to rice monoculture.

## 5. Conclusions

Different feeding schedules significantly affected the growth of carp and tilapia. Feeding five days per week, twice per day in the morning and evening (T10), was observed to be the optimal feeding schedule for both carp and tilapia in the rice field, which also resulted in the highest economic production. Therefore, at the farm level, use of an integrated rice-fish system with carp and tilapia

appears to be an economically viable alternative to rice monoculture when following a feeding schedule of five days per week, twice per day in the morning and evening.

**Author Contributions:** Conceptualization, M.K.U. and M.S.; methodology, M.M.B. and M.P.A.; formal analysis, M.M.B. and M.Z.B.H.; investigation, M.M.B. and M.Z.B.H.; data curation, I.T. and M.Y.A.S.; writing—original draft preparation, M.M.B. and M.K.U.; writing—review and editing, M.S., I.T., A.N.A.H.; supervision, M.K.U. All authors involved in this study helped in writing and improvement of the paper. All authors have read and agreed to the published version of the manuscript.

**Funding:** Universiti Putra Malaysia, Selangor Darul Ehsan, Malaysia for the research facilities.

**Acknowledgments:** The authors are grateful to the "National Agricultural Technology Program-Phase II, Bangladesh Agricultural Research Council" and the Universiti Putra Malaysia, Selangor Darul Ehsan, Malaysia for the research facilities.

**Conflicts of Interest:** The authors declare no conflict of interest.

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
