# Peer review of "Impact of Feeding Schedule on the Growth Performances of Tilapia, Common Carp, and Rice Yield in an Integrated Rice-Fish Farming System"

_sustainability, doi:10.3390/su12208658_

Round 1

Reviewer 1 Report

Please see attached file; twice my comments were lost during saving the draft.

Author Response

Dear Sir,

All the comments made on the file are revised accordingly and marked red throughout the text. Please see the attachment.

Reviewer 2 Report

This study investigated the rice-fish integrated culture system to culture tilapia and carp.

  1. line 22 Authors mentioned “only rice was in the control treatment. This sentence cannot explain the experimental condition which rice was planted in the ponds. It should be revised.
  2. line 24 four days per week in T4
  3. line 28 Fish cannot be planted.
  4. line 42 Therefore, “it” is
  5. line 46 When juvenile Korean rockfish were fed daily is seems as ----- . It is not a sentence.
  6. line 49-51 In case of excessive feeding resulting the poor ----- culture system. It is not a sentence.
  7. Authors need to provide the area of the experiment ponds and how many fish authors used. How to plant the rice field? How depth of the ponds. Authors need to prove more details for this experiment.
  8. Table 1 indicated exp.1 and exp. 2. What are they? What is 75% RF + 5 t/h of compost?
  9. What is BRRI dhan29?
  10. The data of plant height and No. of total tillers hill-1 in Table 5 are different from those of Tables 3 and 4. Why? Need to explain
  11. Table 6 shows the data of growth performance of fish in Page 12. The second paragraph of Page 12 mentioned the growth performance of fish first and then mentioned plant height. It is weird.
  12. Phytoplankton and zooplankton communities are mentioned in Table 8. What are they and their units?
  13. Weed species was measured and presented in Table 9. Why did authors provide weed data in this study? Need to explain in the materials and methods.
  14. Authors mentioned “The feeding frequency has a significant effect on the growth performance of fish.” I do not agree this issue. As we can see that carp fed with rice or floating feed in different feeding frequency, they showed no significant difference.

15, Authors did not discuss the plant harvest and cost benefit in the discussion part.

  1. Lots of grammar error in this article. There are many instances of badly constructed sentences and many spelling and solecism erroe.

Author Response

Response to the comments of the reviewer-2 comments

This study investigated the rice-fish integrated culture system to culture tilapia and carp.

  1. line 22 Authors mentioned “only rice was in the control treatment. This sentence cannot explain the experimental condition which rice was planted in the ponds. It should be revised.

It has been revised accordingly.

  1. line 24 four days per week in T4

It has been added accordingly.

  1. line 28 Fish cannot be planted.

It has been revised accordingly.

  1. line 42 Therefore, “it” is

It has been revised accordingly.

  1. line 46 When juvenile Korean rockfish were fed daily is seems as -----. It is not a sentence.

It has been revised accordingly.

  1. line 49-51 In case of excessive feeding resulting the poor ----- culture system. It is not a sentence.

It has been revised accordingly.

  1. Authors need to provide the area of the experiment ponds and how many fish authors used. How to plant the rice field? How depth of the ponds. Authors need to prove more details for this experiment.

We have mentioned the area of the experiment plots in the revised manuscript.

  1. Table 1 indicated exp.1 and exp. 2. What are they? What is 75% RF + 5 t/h of compost?

Table 1 has been corrected to avoid confusion

  1. What is BRRI dhan29?

It is a rice variety kept in within bracket to avoid confusion

  1. The data of plant height and No. of total tillers hill-1in Table 5 are different from those of Tables 3 and 4. Why? Need to explain

Table 5 has been corrected to avoid confusion

  1. Table 6 shows the data of the growth performance of fish in Page 12. The second paragraph of Page 12 mentioned the growth performance of fish first and then mentioned plant height. It is weird.

We have reorganized to avoid confusion and maintain chronology accordingly

  1. Phytoplankton and zooplankton communities are mentioned in Table 8. What are they and their units?

Table 8 has been deleted to avoid confusion

  1. Weed species was measured and presented in Table 9. Why did the authors provide weed data in this study? Need to explain in the materials and methods.

Table 8 has been deleted to avoid confusion

  1. Authors mentioned, “The feeding frequency has a significant effect on the growth performance of fish.” I do not agree this issue. As we can see that carp fed with rice or floating feed in different feeding frequency, they showed no significant difference.

Thanks for the comment. However, it is very normal that fish growth increase with the increase of feed amount as their required level.

  1. Authors did not discuss the plant harvest and cost-benefit in the discussion part.

We have added this in the discussion part accordingly

  1. Lots of grammar error in this article. There are many instances of badly constructed sentences and many spelling and solecism error.

We revised the language and write-up following the suggestion

Round 2

Reviewer 1 Report

Dear authors

The style and grammar of English are still insufficient and leads to many unclear sentences. I stopped halfway the Introduction, then checked the Conclusion and found identical issues.

Moreover, the rational in the Introduction is not presented logically. I hardly can image all authors had a serious look (proofread) at this revision, which they should, before sending a manuscript to an experienced English editor.

Success with the revision, if the editor allows you to revise, because I am afraid you might have missed a good opportunity.

Author Response

Dear authors

The style and grammar of English are still insufficient and leads to many unclear sentences. I stopped halfway the Introduction, then checked the Conclusion and found identical issues.

Moreover, the rational in the Introduction is not presented logically. I hardly can image all authors had a serious look (proofread) at this revision, which they should, before sending a manuscript to an experienced English editor.

Success with the revision, if the editor allows you to revise, because I am afraid you might have missed a good opportunity.

We revised the English language by MDPI English Editing Service(editing ID: english-22591)

Reviewer 2 Report

  1. Line 22-15 It is not a sentence.
  2. Line 25 fed by --- fed with an artificial
  3. Line 48-50 It is not a sentence.
  4. Line 79 --- fed with an
  5. Line 95 ---- was recorded
  6. Line 100 Authors did mentioned weight gain percentage in the materials and methods. However, we cannot find weight gain percentage data in Table 3.
  7. Line 150-156 The data representation for Table 3 in the content should be revised. I suggest authors to compare data with significant levels.
  8. How many grams of rice bran or artificial floating feed were used in this study?
  9. How to measure CO2 in this study?
  10. Authors need to provide the analysis methods for NH3-N and NO2-N.
  11. From Table 2, we know that authors provided a rice bran for T1 to T5 and an artificial floating diet for T5 to T10. However, we can find T1 to T10 used both rice bran and artificial floating diet for economic analysis in Table 8. Therefore, I can say that the economic analysis data in this study is complete wrong.
  12. I think that the nutrition value of artificial diet is better than that of rice bran. It is reasonable to predict that fish fed with rice bran showed significant lower growth performance than those fed artificial diet in this study. The feeding strategy of diet fed fish also showed the deceasing or increasing trend. However, we find that the feeding strategy of diet fed common carp showed no significant difference. On the other hand, tilapia showed significant difference. Authors need to discuss them why and how.

Author Response

We revised the English language by MDPI English Editing Service(editing ID: english-22591)

 Comments and Suggestions for Authors

  1. Line 22-15 It is not a sentence.

It has been revised accordingly

  1. Line 25 fed by --- fed with an artificial

It has been revised accordingly

  1. Line 48-50 It is not a sentence.

It has been revised accordingly

  1. Line 79 --- fed with an

It has been revised accordingly

  1. Line 95 ---- was recorded

It has been revised accordingly

  1. Line 100 Authors did mention weight gain percentage in the materials and methods. However, we cannot find weight gain percentage data in Table 3.

It has been revised accordingly

  1. Line 150-156 The data representation for Table 3 in the content should be revised. I suggest authors to compare data with significant levels.

It has been revised accordingly

  1. How many grams of rice bran or artificial floating feed were used in this study?

It has been mentioned as % body weight of fish

  1. How to measure CO2in this study?

It has been mentioned accordingly

Round 3

Reviewer 1 Report

Dear authors

Thank you for the major edits in the English. Somehow various spaces between words were lost in the pdf ( I only marked those on the first pages, but noticed these also on other pages). I have still some sparse comments on the wording and style; see attached: lines 48, 67-68, 153-155, 168-169, 208-209.

In my opinion, the paragraph on lines 248 to 253 is superfluous for your discussion because not directly related to your results and reads like a generic review on the topic; I suggest to delete those. 

Success with the last changes

Author Response

Response to Reviewer 1 Comments

Point 1: Thank you for the major edits in the English. Somehow various spaces between words were lost in the pdf ( I only marked those on the first pages, but noticed these also on other pages). I have still some sparse comments on the wording and style; see attached: lines 48, 67-68, 153-155, 168-169, 208-209.

Response 1: Thanks for the comments. We have revised the spaces between the words. We checked these problems in the revised manuscript.

Point 2: In my opinion, the paragraph on lines 248 to 253 is superfluous for your discussion because not directly related to your results and reads like a generic review on the topic; I suggest to delete those. 

Response 2: The paragraph has been deleted as suggested. 

Reviewer 2 Report

  1. In Table 3

I think the yield data has something wrong.

1 ha = 10000 m2

For example, the final weight of tilapia is 147.3 g in T1 treatment.

Then, 147.3 g x 3 fish/m2 x 10000 = 4419000 g = 4419 kg

4419 kg x 0.91 (survival) = 4021 kg

The yield data of T1 treatment should be 4021 kg instead of 4462.2 kg. Authors need to check the data of Table 3.

  1. Line 211-212 Authors mentioned “The net return from rice–fish culture was significantly higher than in the control (T0) when compared with the revenue generated from the plot of rice–fish farming due to the presence of fish”. However, I did not see any significant sign for net return in Table 3.
  2. What is BCR?

Author Response

Response to Reviewer 2 Comments

Point 1: In Table 3

I think the yield data has something wrong.

1 ha = 10000 m2

For example, the final weight of tilapia is 147.3 g in T1 treatment.

Then, 147.3 g x 3 fish/m2 x 10000 = 4419000 g = 4419 kg

4419 kg x 0.91 (survival) = 4021 kg

The yield data of T1 treatment should be 4021 kg instead of 4462.2 kg. Authors need to check the data of Table 3.

Response 1: Thank you for raising this important point. We revised the Table 3 accordingly and made changes throughout the manuscript (Table 8, lines 31, 156, 216).

Point 2: Line 211-212 Authors mentioned “The net return from rice–fish culture was significantly higher than in the control (T0) when compared with the revenue generated from the plot of rice–fish farming due to the presence of fish”. However, I did not see any significant sign for net return in Table 3.

Response 2: The net returns are shown in Table 8. We have revised the sentence to avoid confusion (lines 213-215).

Point 3: What is BCR?

Response 3: BCR = benefit–cost ratio